# Patient-Derived Orthotopic Xenograft (PDOX) Mouse Models of Primary and Recurrent Meningioma

**DOI:** 10.3390/cancers12061478

**Published:** 2020-06-05

**Authors:** Huiyuan Zhang, Lin Qi, Yuchen Du, L. Frank Huang, Frank K. Braun, Mari Kogiso, Yanling Zhao, Can Li, Holly Lindsay, Sibo Zhao, Sarah G. Injac, Patricia A. Baxter, Jack M. Su, Clifford Stephan, Charles Keller, Kent A. Heck, Akdes Harmanci, Arif O. Harmanci, Jianhua Yang, Tiemo J. Klisch, Xiao-Nan Li, Akash J. Patel

**Affiliations:** 1Laboratory of Molecular Neuro-Oncology, Department of Pediatrics, Preclinical Neuro-Oncology Research Program, Baylor College of Medicine, Houston, TX 77030, USA; hxzhang3@texaschildrens.org (H.Z.); lqi@luriechildrens.org (L.Q.); yuchen.du@northwestern.edu (Y.D.); fkbraun@outlook.com (F.K.B.); mxkogiso@texaschildrens.org (M.K.); hblindsa@texaschildrens.org (H.L.); sibo.zhao@cookchildrens.org (S.Z.); sginjac@texaschildrens.org (S.G.I.); pabaxter@texaschildrens.org (P.A.B.); 2Department of Pediatrics, Texas Children’s Cancer Center, Texas Children’s Hospital, Houston, TX 77030, USA; yxzhao2@texaschildrens.org (Y.Z.); msu@bcm.edu (J.M.S.); jianhuay@bcm.edu (J.Y.); 3Program of Precision Medicine PDOX Modeling of Pediatric Tumors, Ann and Robert H. Lurie Children’s Hospital of Chicago and Department of Pediatrics, Northwestern University Feinberg School of Medicine, Chicago, IL 60611, USA; 4Division of Experimental Hematology and Cancer Biology, Brain Tumor Center, Cincinnati Children’s Hospital Medical Center, Cincinnati, OH 45229, USA; frank.huang@cchmc.org; 5Department of Pediatrics, College of Medicine, University of Cincinnati, Cincinnati, OH 45221, USA; 6Institute of Biosciences and Technology, Texas A&M Health Science Center, Houston, TX 77030, USA; cli@medicine.tamhsc.edu (C.L.); cstephan@ibt.tamhsc.edu (C.S.); 7Children’s Cancer Therapy Development Institute, Beaverton, OR 97005, USA; charles@cc-tdi.org; 8Department of Pathology, Baylor College of Medicine, Houston, TX 77030, USA; heck@bcm.edu; 9Center for Computational Systems Medicine, School of Biomedical Informatics, University of Texas Health Science Center at Houston, Houston, TX 77030, USA; akdes.harmanci@uth.tmc.edu; 10Center for Precision Health, School of Biomedical Informatics, University of Texas Health Science Center at Houston, Houston, TX 77030, USA; arif.o.harmanci@uth.tmc.edu; 11Jan and Duncan Neurological Research Institute, Texas Children’s Hospital, Houston, TX 77030, USA; tiemo.klisch@bcm.edu; 12Department of Neurosurgery, Baylor College of Medicine, Houston, TX 77030, USA

**Keywords:** meningioma, xenograft, PDOX, modelling, HDAC inhibitor

## Abstract

Background. Meningiomas constitute one-third of all primary brain tumors. Although typically benign, about 20% of these tumors recur despite surgery and radiation, and may ultimately prove fatal. There are currently no effective chemotherapies for meningioma. We, therefore, set out to develop patient-derived orthotopic xenograft (PDOX) mouse models of human meningioma using tumor. Method. Of nine patients, four had World Health Organization (WHO) grade I tumors, five had WHO grade II tumors, and in this second group two patients also had recurrent (WHO grade III) meningioma. We also classified the tumors according to our recently developed molecular classification system (Types A, B, and C, with C being the most aggressive). We transplanted all 11 surgical samples into the skull base of immunodeficient (SCID) mice. Only the primary and recurrent tumor cells from one patient—both molecular Type C, despite being WHO grades II and III, respectively—led to the formation of meningioma in the resulting mouse models. We characterized the xenografts by histopathology and RNA-seq and compared them with the original tumors. We performed an in vitro drug screen using 60 anti-cancer drugs followed by in vivo validation. Results. The PDOX models established from the primary and recurrent tumors from patient K29 (K29P-PDOX and K29R-PDOX, respectively) replicated the histopathology and key gene expression profiles of the original samples. Although these xenografts could not be subtransplanted, the cryopreserved primary tumor cells were able to reliably generate PDOX tumors. Drug screening in K29P and K29R tumor cell lines revealed eight compounds that were active on both tumors, including three histone deacetylase (HDAC) inhibitors. We tested the HDAC inhibitor Panobinostat in K29R-PDOX mice, and it significantly prolonged mouse survival (*p* < 0.05) by inducing histone H3 acetylation and apoptosis. Conclusion. Meningiomas are not very amenable to PDOX modeling, for reasons that remain unclear. Yet at least some of the most malignant tumors can be modeled, and cryopreserved primary tumor cells can create large panels of tumors that can be used for preclinical drug testing.

## 1. Introduction

Meningiomas constitute 36% of all primary brain tumors [1,2]. While the majority of meningiomas are benign (World Health Organization, WHO, grade I) and can be cured surgically, approximately 15–20% of meningiomas are atypical or malignant (grade II and III, respectively) [3,4,5,6]. Despite surgical or radiation therapy, approximately 41% of these tumors recur in five years. There are currently no effective therapies for patients with surgery- and radiation-refractory high-grade meningiomas. 

One major challenge to developing chemotherapies for atypical and malignant meningioma is the scarcity of clinically relevant and reliable models. Studies using in vitro cultured meningioma cells are hampered by the fact that primary human meningioma cells are slow-growing and enter senescence after five to seven passages. Thus, cultured cell studies rely on either early-passage primary tumor cells [7,8] or immortalized meningioma cells [7,8,9,10,11,12]. Immortalized cell lines have their own disadvantage, in that the introduction of genetic changes that allow perpetual division also prevents the cells from being molecularly identical to the tumor being modeled. In light of recent studies finding genomically distinct subclasses of meningiomas [3,4,13,14,15,16,17], it is important to develop models that enable us to dissect the consequences of these molecular profiles. 

A more physiologically accurate way to mimic tumor characteristics is to implant patient tumor cells directly into immunodeficient mice. This strategy has worked with many other tumor types [18,19,20], and we and others have shown that these patient derived orthotopic xenograft (PDOX) models replicate the biology and behavior of the original brain tumors. In most cases, the patient’s tumor cells are transformed into cell lines that are then injected into the animals [21,22,23,24,25,26,27,28]. For example, Nigim et al. [28] attempted to engraft malignant meningioma cells cultured in fetal calf serum, but the cells did not produce a sizeable tumor mass, so the authors developed tumorsphere cultures instead and transplanted those cells into frontal subdural of severe combined immunodeficiency (SCID) mice, with success. 

In this report, we present our efforts in developing PDOX models from fresh surgical specimens of all three WHO grades of meningiomas, taken from nine different patients. Only two tumors successfully engrafted, and they were both from the same patient (a primary and a recurrent tumor). Using RNA sequencing, we were able to demonstrate that the xenografts recapitulated the genomic features of the original tumors. Interestingly, our xenografts could not be subtransplanted or passaged; however, using primary tumor cells, we were able to consistently create new models. We conducted a preclinical drug screen to identify candidate therapies and identified that Panobinostat, a histone deacetylase (HDAC) inhibitor, prolonged animal survival. 

## 2. Results

### 2.1. PDOX Models Derived from Primary and Recurrent Meningiomas 

Our experience here conforms with that of a previous attempt to transplant fresh tumor cells into mice [28]: Meningiomas do not “take” easily in mice. Only two of 11 tumors from nine patients produced tumors in the PDOX mice. Notably, both tumor-forming meningiomas were derived from the same patient (K29), who had had resection of a WHO grade II primary tumor and a WHO grade III recurrent tumor. The primary tumor, designated K29P, “took” in seven of 10 mice transplanted, and the recurrent tumor, designated K29R, took in one of four mice transplanted (Table 1). These PDOX models were thus designated K29P and K29R. 

Cross-sections of whole mouse brains (Figure 1A) and small animal magnetic resonance imaging (MRI) revealed large skull base meningiomas with obstructive hydrocephalus (partially enlarged lateral ventricles caused by restricted cerebrospinal fluid (CSF) circulation) (Figure 1B). Log-rank analysis showed significantly shorter survival in the recurrent model (K29R-PDOX) than in the primary model (K29P-PDOX) (Hazard ratio = 2.474; *p* = 0.0390, Figure 1C), as consistent with the more aggressive behavior expected of a WHO grade III tumor. 

### 2.2. PDOX Models Recapitulate the Histopathologic Features of the Patient Tumors

To determine if the xenograft tumors replicate the histopathologic characteristics of the original patient tumors, we performed standard hematoxylin and eosin (H&E) and immunohistochemistry (IHC) staining on paraffin sections of patient and matching xenograft tumors. The original tumors as well as their xenograft counterparts were fibroblastic with densely cellular foci and focal “whorl formations” by spindle-shaped tumor cells (Figure 1D) with nuclear atypia, high nuclear/cytoplasmic ratio, and focal areas of necrosis. The xenografts showed irregular protrusions into the adjacent brain, which is consistent with brain invasion, along with scattered mitotic figures, both of which are features of high-grade meningiomas.

To confirm the human origin of the PDOX tumors, we stained the patient and xenograft slides with human-specific antibodies against mitochondria (MT), Ki-67, and vimentin (Figure 1E). Strong mitochondrial positivity was detected in nearly all the xenograft as well as patient tumor cells. Cell proliferation rate (Ki-67) was 2–5% in the primary tumor and xenografts but rose to ~10% in the recurrent tumor and xenografts. Expression of markers of cell proliferation (Ki-67), the neurofilament vimentin (VIM), blood vessel growth (von Willebrand factor, VWF), and platelet-derived growth factor (PDGFR1) were similar between the human and the matching PDOX tumors (Figure 1E).

### 2.3. PDOX Models Can also be Generated from Cryopreserved Cells

Since all tumor-bearing mice died of intracranial meningioma, we set out to generate a sustainable supply of PDOX models from K029P and K029R by serial transplantation [29,30]. After harvesting passage I xenograft tumors and implanting them intracranially into new recipients (now PDOX-II, 1 × 10^5^ cells/mouse, 10 mice per model), we monitored the mice for up to 12 months but found no tumor formation in any of the recipient mice. This experiment was repeated three times from K29P-PDOX-II and twice with K29R-PDOX-II by using xenograft cells from different donor mice. No tumor formation was seen in any of the 50 mice. 

Another way to achieve a continuous supply of PDOX models is to use original tumor cells. Since our PDOX cells failed to repopulate new xenografts following subtransplantation, we asked whether our K29 cryopreserved samples remained tumorigenic. Indeed, we had cryopreserved a large number of primary cells of K29P (1.8 × 10^7^ cells) and K29R (2 × 10^7^ cells) from the original tumors. As shown in Figure 2A, tumor take rates (at 1 × 10^5^ cells/mouse) were maintained for K29P in 3/3 mice (100%) after 22 months (679 days) in liquid nitrogen and for K29R at 5/5 and 4/4 mice (100%) after approximately three years (1044 and 1057 days, respectively) in cryopreservation. Survival times for these animals (~240 days in the primary and ~160 days in the recurrent PDOX mice) did not differ from that of the animals that received fresh surgical tumors (*p* > 0.05) (Figure 2A). These data confirm that a relatively large cohort of tumor-bearing mice can be generated from existing original, cryopreserved meningioma tumor cells, if adequate starting material has been stored. 

To determine whether transplanting larger numbers of cells would expediate tumor growth so as to save time for preclinical drug testing, we doubled the number of injected tumor cells to 2 × 10^5^ cells/mouse. This reduced the median K29P survival times from 272 to 246 days (9.5% reduction) (Figure 2B). Further increasing injected cell numbers (to 4 × 10^5^ cells) only reduced the median survival to 243 days (or just a 1.1% reduction) (*p* > 0.05). The recurrent tumor K29R had a median survival of 201 days for 1 × 10^5^ cells/mouse, which was reduced to 154 days with 2 × 10^5^ cells (23.4% reduction) and 167 days with 4 × 10^5^ cells (16.9% reduction)—clearly, using more tumor cells is not necessarily better, even for the most aggressive tumors (Figure 2B). These data suggest that 10,000 initial tumor cells are sufficient to generate a PDOX model of a patient tumor, if it will take. Given the limited supply of patient tumor cells, it is equally important to note that using more tumor cells for PDOX models can be counterproductive. 

### 2.4. Gene Expression Profiles Were Maintained in the PDOX Tumors

To further characterize the two PDOX models and perhaps to uncover molecular determinants that influence engraftment, we compared the gene expression profiles of the patient’s original tumors (K29P and K29R) and the xenografts (K29P-PDOX and K29R-PDOX) using RNA sequencing (RNA-seq). The original tumors had been previously profiled using RNA-seq [31] and found to be Type C tumors, the most aggressive type. Using a random forest classifier model [31] confirmed that both PDOX models were also classified as Type C tumors, further confirming that the original tumors maintained their molecular identity, even after transplantation in mice. 

We compared copy number alterations between the two original tumors, the derived PDOX models as well as two unrelated control tissues (see below) using the CaSpER method, used for detection of copy number variants using RNA-seq [32]. The PDOX tumors retained all of the copy-number variations seen in the original tumors (Figure 2C). Because it is difficult to acquire matching normal tissue for meningioma (meningothelial/arachnoidal cells), we included two normal cerebral tissues obtained from warm autopsy as references (<6 hr postmortem). A total of 2973 genes were differentially expressed in the meningioma samples compared to the normal tissue (*p* < 0.05 and false discovery rate (FDR) < 0.05, Figure 2D). However, compared with the originals, the primary and recurrent xenografts exhibited a very high correlation coefficient (*r* = 0.99 for both K29P and K29R, Figure 2E). 

We reasoned that the inability of the PDOX tumors to be subtransplanted may result from slight differences between the original tumors and the xenografts. Comparing the xenograft’s gene expression with the gene expression of the primary tumors of patient 29, we could identify only three genes that were significantly increased (*POTEF*, *ADCY2*, and *LEPR*) but 43 genes that were dramatically down-regulated (Figure 2F). Among the down-regulated genes were 15 that have been reported to have pro-proliferative effects, such as *TIE1*, *TBX2*, *ARHGEF15*, *S100A8*, and *FCN1* [33,34,35,36,37,38]. Eleven of the genes have been reported to be down-regulated by the tumor suppressor TCF21, such as *FLI1*, *SEMA3G*, and *PCDH12* [39,40]. We also noticed that 10 genes were associated with the VEGF-A complex, such as *FLT1*, *VWF*, *MMP9*, and *ESAM* [41]. Looking specifically at the expression of vascular endothelial growth factor (VEGF) family members (A, B, C, and D), we discovered that the xenograft tumors expressed only one-seventh of the levels of VEGF-C found in the primary tumor. Given the critical role of VEGF-mediated signaling in tumor initiation, proliferation, and migration [42], this observation warrants further studies to see whether interfering with this pathway could restrain aggressive meningioma behavior. Furthermore, since VEGF receptor 1 plays important roles in tumor neogenesis, this finding provides one possible explanation for the failure of subtransplanted xenograft cells to generate a new tumor in mouse brains.

### 2.5. High-Throughput in Vitro Chemical Screen Identify HDAC Inhibitors 

We showed that the PDOX tumors recapitulate to a high degree the transcriptional profiles of the original tumor tissue, with just 46 genes being differentially expressed between the original tumor and the PDOX tumors. Because the recurrence (K29R) grew faster than the original tumor (K29P), we used an in vitro screening approach using the K29R primary cells. We incubated the primary meningioma cells in standard fetal bovine serum (FBS)-based media in which the meningioma cells grew as monolayer with a fibroblast-type appearance (Figure 3A). By plating cells on low-attaching plates with *EGF* and basic *FGF* supplements, we could also establish tumorspheres (Figure 3A). For ease of use, we used the monolayer cells, and screened a panel of 60 drugs that were previously selected by a group of clinical neuro-oncologists and had been successfully tested in preclinical settings in diffuse intrinsic pontine glioma (DIPG) [43] (Table 2). We examined tumor cell proliferation after seven days of exposure to the compounds. Although the majority of the drugs (48/60, or 74%) were not active in any scenario (Figure 3B,C), eight drugs were active in both the primary tumor (K29P-PC) and recurrent tumor (K29R-PC) primary cultures (PC): Three HDAC inhibitors (Panobinostat, SAHA, and Eninostat), two cyclin-dependent kinase inhibitors (Dinaciclib and BMS-387032), and inhibitors of PI3K/mTOR (BEZ 235), heat shock protein 90 (AUY922), and proteasome (Cafilzomib) (Figure 3C and Figure 4). Although the transcriptional profiles of the primary and recurrent tumors were very similar, we identified two drugs (MLN8237 and BIX 01294) that were active only in K29P-PC, and two different drugs (Flavopiridol and Dasatinib) that were active only in the recurrent K29P-PC (Figure 3C). These data shed light on possible entry points to influence meningioma biology as well as tumor-specific behavior. 

### 2.6. Panobinostat Significantly Prolonged Animal Survival in the Recurrent Tumor Model 

Since HDAC inhibition made the strongest showing in the screen, we sought to determine whether this could provide an approach to future medications in vivo. We decided to focus on Panobinostat, which was recently shown to be able to penetrate the blood–brain barrier (BBB) and has entered phase I clinical trials [43]. Since the K29R tumors were less responsive than K29P in vitro (Figure 5), we focused our in vivo efficacy analysis on K29R-PDOX mice. The tumor cells were implanted intracranially (10 mice per group, vehicle-only or Panobinostat treatment; see methods) and allowed to grow for two weeks to form solid tumors. After two weeks, Panobinostat, either treatment or vehicle, was administered by intraperitoneal injection (10 mg/kg/day, in two cycles of five days on and five days off). Animals were monitored daily for head tilt, gait abnormalities, or weight loss, at which point the animals were euthanized. Whereas vehicle-treated mice were euthanized around 154 days, treated mice survived 258 days (>60% longer) (*p* = 0.039) (Figure 5A). 

To understand the mechanism of action, we incubated K29R cells with Panobinostat for 24 and 48 hours at 0, 0.1, and 0.5 µM doses. Western blot analysis revealed high expression of HADC1 and HDAC2, the molecular targets of Panobinostat, and a significant increase of acetylated histone H3 that started at 24 hours and peaked at 48 hours (for 0.5 µM). The increase of acetyl-histone H3 was accompanied by cleaved Poly (ADP-ribose) polymerase (PARP), indicating the induction of apoptosis, although the cleavage of caspase 3 was much less prominent. The expression of Enhancer of zeste homolog 2 (EZH2) was not significantly altered (Figure 5B and Appendix A). Altogether, these proof-of-principle data support the role of HDAC inhibition and apoptosis induction in mediating Panobinostat-induced cell death, similar to previous reports [43].

## 3. Discussion

Given the lack of intracranial PDOX meningioma models and the difficulties others have encountered in attempting them [28], we set out to determine if direct intracranial implantation of primary meningioma tumor cells from a range of patient tumor types would allow us to develop patient-specific models that recapitulate the features of the original tumor. Unfortunately, the meningioma resisted our best efforts: Out of 11 tumor samples from nine patients, only two tumor samples reproduced meningioma in mice. Although these samples were classified as WHO grade II for the primary tumor and WHO grade III for the recurrence, both tumor samples were classified as Type C tumors in our molecular classification system. It is tempting to conclude that only Type C tumors are amenable to PDOX modeling—if it were not for the fact that three other Type C tumors (also a mix of WHO grades II and III, Table 1) also failed to take. The most we can say, then, is that some malignant (Type C) meningioma can be modeled in a PDOX system.

Nevertheless, detailed characterization confirmed that the two successful PDOX models faithfully replicated the histopathological features and transcriptional profiles of the original patient tumors. Through these models we were able to identify a subset of genes that were differentially expressed between the primary and the recurrent tumors. Although these two PDOX models were not subtransplantable, the tumor cells survived long-term cryopreservation and were able to provide a sustained supply of PDOX models. We were also able to examine the in vitro antitumor activities of 60 drugs and subsequently test one drug in vivo. The fact that Panobinostat, an HDAC inhibitor, prolonged survival accords with recent studies showing that there is an epigenetic component to meningioma biology [3,14,17].

Low-grade brain tumors of various types have been shown previously to have significantly decreased tumor take rate [44,45]. In pediatric low-grade gliomas, for example, we were able to develop only one model from 25 implanted tumors [45]. Similar findings have been reported in other types of human cancers [44,46]. Unlike many high-grade brain tumors, however, our newly established PDOX models of meningioma were not able to repopulate a new xenograft despite repeated tests. The failure of our four grade I meningioma samples to take in any of 30 mice, and the failure of repopulation attempts, suggests that this type of tumor is not amenable to PDOX modeling. Nevertheless, before drawing any firm conclusions, the field needs a systematic analysis of the genomic differences between tumor-forming and non-tumorigenic meningiomas to better understand why the vast majority of these tumors do not grow as xenografts. 

Despite the difficulties of creating the PDOX models, they are extremely useful for preclinical drug testing. We found that the majority (75%) of drugs effective in the primary tumor remained effective in the recurrent tumor. While we might anticipate a drop-off in efficacy of primary tumor-responsive drugs in the recurrent tumor, it was a pleasant surprise to discover two drugs that converted from inactive in the primary tumor to active in the recurrent tumor. PDOX models could thus reveal important characteristics of individual patient tumors, serving the goals of personalized medicine, but the generation of such models needs to become more reliable and cost effective for them to be more broadly useful.

## 4. Materials and Methods 

### 4.1. Patient Tumor Tissue Specimen 

We acquired freshly resected meningioma tumor specimens, over a period of 2 years, from nine patients undergoing surgery at Baylor College of Medicine (BCM) after the patients gave written, informed consent following a protocol approved by the Institutional Review Board (IRB), H-35355 (approved on 31 July 2014) and H-14435 (initially approved on 16 September 2003). Animal studies were conducted under a protocol approved by the Institutional Animal Care and Use Committee (IACUC) of BCM, protocol AN-4548, initially approved on 23 July 2013. Patient demographics and clinical information are presented in Table 1. Two of the nine patients had surgical removal of both primary and recurrent tumors, bringing the total number of tumor samples to 11: Four WHO grade I, five WHO grade II, and two WHO grade III. All samples were analyzed histopathological by a neuropathologist (K.A.H.) and graded following the 2016 WHO guidelines [47]. Ten of the 11 tumors were subtyped according to our recently published expression-based classification system [3].

Tumor tissues were processed following our established protocol. Briefly, tumor tissues were washed and minced with fine scissors into small fragments. Single cells and small clumps of three to five cells were collected with a 40 μm cell strainer and re-suspended in (Dulbecco’s Modified Eagle Medium (DMEM) growth medium to achieve a final concentration of 1 × 10^6^ live cells per mL, as assessed by trypan blue staining. 

### 4.2. Long-Term Storage of Xenograft Cells in Liquid Nitrogen 

As described previously [43,47,48,49], a tumor cell suspension was centrifuged at 200× *g* for 10 min. The cell pellet was re-suspended with DMEM medium supplemented with 10% fetal bovine serum and 10% dimethyl sulfoxide (DMSO), then aliquoted into cryovials at 10^6^ live cells /mL. After having been kept at −80 °C overnight, the cryovials were transferred into liquid nitrogen for long-term storage. To retrieve cells for inoculation into SCID mice, the cells were quickly thawed in a 37 °C water bath, washed once with normal growth medium, and counted with trypan blue before being used for intracranial implantations.

### 4.3. Direct Hetero-Transplantation of Primary Tumor Cells into Mouse Brain 

Rag2 SCID mice [49] were bred and housed in a pathogen-free animal facility at Texas Children’s Hospital, Houston, TX. All experiments were conducted using an Institutional Animal Care and Use Committee (IACUC)-approved protocol. Surgical transplantation of tumor cells into the skull base of the mouse was performed using a free hand implantation strategy as described previously [29,30]. Although some of the tumors originated in the convexity, we injected tumor cells to the skull base to prevent back-flow of the cell suspensions. Mice were assessed for tumor formation over the next 12 months. 

In brief, male and female mice, aged 12–16 weeks, were anesthetized with inhaled isoflurane or intraperitoneal injection of pentobarbital (50 mg/kg). Tumor cells (1 × 10^5^) were suspended in 2 µL of culture medium and injected via a 10 µL 26-gauge Hamilton Gastight 1701 syringe needle into the skull base of the right frontal/temporal region (2 mm to the right of the midline, 2.5 mm anterior to the lamboidal suture, and 7–8 mm deep, where the tip of the needle touches the bone of the cranial base). For each tumor, up to 10 mice were implanted (see Table 1).

The mice were visually inspected daily for neurological deficits, such as head tilt, paralysis, decreased mobility, gait abnormalities, or weight loss. Upon development of any of these symptoms, the mice were euthanized and their brains removed for histopathologic examination. If the mice showed no deficits, they were euthanized and examined at one year of age. 

For serial subtransplantation of xenografts, tumors from donor mice (Passage I, P-I) were dissected, mechanically dissociated, and injected into brains of recipient SCID mice (Jackson Lab, Sacramento, CA) at 1 × 10^5^ cells/mouse as described above. For long-term cryopreservation, freshly prepared patient and xenograft tumor cells were resuspended with DMEM medium supplemented with 10% fetal bovine serum and 10% dimethyl sulfoxide; after overnight incubation at −80 °C, the cells were stored in liquid nitrogen. For retransplantation into SCID mice, tumor cells were quickly thawed at 37 °C, washed, counted with trypan blue, and injected intracranially (five mice per tumor, 1 × 10^5^ cells per injection) as described above. 

### 4.4. Immunohistochemical (IHC) Staining and Western Blot Analysis

IHC staining was performed using a Vectastain Elite kit (Vector Laboratories, Burlingame, CA, USA) as described previously [30]. In brief, we used a microwave antigen retrieval method with 30 mM sodium citrate buffer to enhance immunostaining. Endogenous peroxidase was quenched using hydrogen peroxide before sections were blocked in Avidin D and a biotin-locking reagent. Primary antibodies included mouse monoclonal antibodies or rabbit polyclonal antibodies against Ki67 (ab833-500, Abcam Inc., Cambridge, UK), Cow Glial Fibrillary Acidic Protein (GFAP, M0761, DAKO Corp. Carpinteria, CA), human-specific mitochondria (MAB1273, Chemicon International Inc. Temecula, CA, USA), p53 (sc-6243, Santa Cruz Biotechnology, Inc. Dallas, TX, USA), Vimentin (VMT, M0725, DAKO Corp. Carpinteria, CA, USA), anti-von Willibrand Factor (VWF, AB7356, Millipore Crop. Burlington, MA, USA), EMA (M0613, DAKO Corp. Carpinteria, CA, USA), D2-40 (M3619, DAKO Corp. Carpinteria, CA, USA), and PDGFR1 (PV3811, Fisher Scientific, Waltham, MA, USA). After incubation with primary antibodies for 90 minutes at room temperature, the slides were probed with the appropriate secondary antibodies (1:200) for 30 minutes, and the final signal was amplified using the 3,3′-diaminobenzidine (DAB) substrate kit. 

For western blot analyses, whole cell extracts were prepared in RIPA (radioimmunoprecipitation assay) buffer plus proteinase inhibitors. Equal amounts of cell extracts were separated by SDS-PAGE (sodium dodecyl sulphate-polyacrylamide gel electrophoresis), transferred to polyvinylidence fluoride (PVDF) membranes (BioRad, Hercules, CA, USA), blocked with 5% BSA (Bovine Serum Albumin) in TBST (Tris-buffered saline, 0.1% Tween 20) for 1 hour at room temperature (25 °C), and probed with appropriate antibodies overnight at 4 °C. The membranes were washed, incubated with secondary antibodies for 1 hour at room temperature, and immunoreactivity was visualized using an ECL-Plus Western Detection Kit (GE Health Care, Buckinghamshire, UK). Glyceraldehyde 3-phosphate dehydrogenase (GAPDH) was used to ensure equal protein loading. Primary antibodies used in this study were: Anti-H3Ac (Abcam, ab47915), anti-GAPDH (Abcam, ab128915), anti-HDAC1 (Cell Signaling 5356T, Danvers, MA, USA), anti-HDAC2 (Cell Signaling, 5113T), anti-PARP (Cell Signaling, 9532S), anti-Caspase 3 (Cell Signaling, 9662S), and anti-EZH2 (Active motif, Carlsbad, CA). Secondary antibodies included horseradish peroxidase (HRP)-linked anti-mouse IgG (Cell Signaling, 7076S) and HRP-linked anti-rabbit IgG (Cell Signaling, 7074S).

### 4.5. RNA-Seq Analysis

RNA-seq libraries for transcriptome analysis were prepared using the TruSeq RNA Sample Preparation Kit (Illumina, San Diego, CA, USA) and Agilent (Santa Clara, CA, USA) Automation NGS system per manufacturers’ instructions. In brief, after enrichment for poly(A) RNA by using oligo dT magnetic beads of 1 µg of total RNA, RNA was fragmented with divalent cations, converted into complementary DNA (cDNA), and were repaired using T4 DNA polymerase, Klenow polymerase, and T4 polynucleotide kinase. A 3’ A-tailing with exo-minus Klenow polymerase was followed by ligation of Illumina paired-end oligo adapters. Ligated DNA was PCR amplified for 15 cycles and purified (AMPure XP bead kit, Beckman Coulter, Indianapolis, IN, USA). Quality and quantity were analyzed using an Agilent Bioanalyzer High Sensitivity chip.

RNA-seq reads were analyzed using FastQC for quality control, aligned to the human reference genome (hg38) using TopHat [50] with default parameters and assessed gene expression using HTseq [31] and DESeq2 [51]. In DESeq2, we modeled the raw read counts to follow a negative binomial distribution, and a generalized linear model [52] was used to fit the raw read counts of each gene.

### 4.6. Small-Animal Magnetic Resonance Imaging

Small-animal MRI studies were performed with a Bruker Biospec 9.4-T horizontal bore MRI scanner (Bruker Corp., Billerica, MA, USA) with a 35-mm inner diameter mouse radiofrequency coil. Animals were anesthetized through inhaled isoflurane and monitored for respiration and with electrocardiogram. Both T1 and T2 images were captured.

### 4.7. Primary Cell Culture Establishment and Propagation

Patient tumor tissue was dissociated and incubated both in 10% fetal bovine serum (Atlanta Biologicals, Inc., Flowery Branch, GA, USA) and DMEM media favoring growth of monolayer tumor cells. The media was changed every three days and split 1:2 when confluent.

### 4.8. High-Throughput Drug Screening 

We used a panel of 60 anti-cancer drugs (Table 2) to screen meningioma cells for drug sensitivity, as described previously [53]; the majority of these compounds are currently in clinical trials. Each compound was tested in four concentrations (10 μM, 1 μM, 100 nM, and 10 nM) in triplicate in a 384-well format. Early-passage cells (i.e., within the first five passages) cultured from PDOX tumors were plated at a density of 4000 cells/well and incubated at 37 °C with humidified 5% CO_2_ for 7 days. Cell viability was measured with the BioTek Synergy 2 (Agilent, Santa Clara, CA, USA). IC50 (half maximal inhibitory concentration) values were determined by a nonlinear best-fit method using Excel Solver (Microsoft Office). We evaluated the effects of drugs on cell viability using the two-sample Kolmogorov–Smirnov test (KS2TEST) comparing each experimental well to the DMSO control. A dose-response curve was generated, and the normalized area under curve (AUC) was calculated by a nonlinear regression analysis using a four-parameter logistic equation (GraphPad Prism, version 5, GraphPad Software). 

### 4.9. Testing Panobinostat Therapy in Mice

Panobinostat (Selleckchem, Houston, TX, USA) was dissolved in DMSO at 150 mg/mL and used as previously described [43]. Since the K29R-PDOX model showed a more aggressive phenotype (greater tumor volume and earlier lethality), we used a new set of K29R-PDOX mice to test the drug. Meningioma tumor cells from K29R were implanted intracranially (1 × 10^5^ cells/mouse) into 20 mice and allowed to grow for two weeks to form solid tumors, 10 for placebo treatment and 10 for Panobinostat treatment. Panobinostat was administered by intraperitoneal injection at 10 mg/kg/day, for two cycles of five days on and five days off. Animals were monitored for changes of body weight and signs of sickness and were euthanized when moribund. Changes of animal survival times between the treated and the control groups were analyzed through log-rank analysis. 

## 5. Conclusions

In conclusion, we showed that, despite low tumor take rate, PDOX models can be established from surgical specimens of grade II atypical and/or grade III anaplastic meningioma tumors. Our novel set of matching pairs of primary and recurrent meningioma models replicated histopathological and genomic features of the original patient tumors. Since the tumor-forming capacity was well maintained in cryopreserved tumor cells, our data highlight the importance of maximizing collection and preservation of viable tumor in liquid nitrogen. Our in vitro and in vivo drug testing demonstrated the utility of these models in the development of new anti-meningioma therapies. It is crucial for future studies to shed light on what it is about human meningioma that makes it so difficult to model in animals.

## Figures and Tables

**Figure 1 cancers-12-01478-f001:**
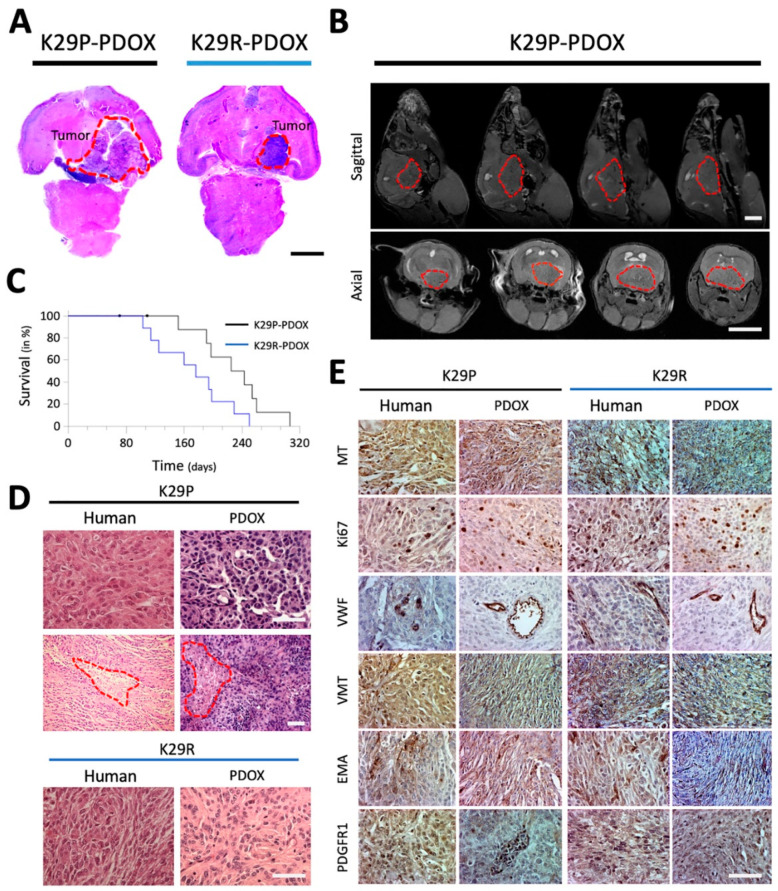
Characteristics of PDOX models of primary and recurrent meningioma. (**A**) H&E-stained paraffin sections of whole mouse brains bearing the intracranial xenograft tumors derived from the primary grade II (K29P) and recurrent grade III (K29R) meningiomas. (**B**) MRI showing the growth of cranial-base xenografts (red circles) of K29P-PDOX and obstructive hydrocephalus. (**C**). Kaplan–Meier survival curves for K29P and K29R PDOX models (*p* = 0.0390, *n* = 8 out of 10 and 9 out of 9, respectively). (**D**). H&E staining of xenograft and human tumors in the K29P and K29R-PDOX models. Areas of necrosis are circled in red. (**E**)**.** Representative images of IHC staining in matching patient tumor and PDOX models. MT: Mitochondria (human-specific). Ki-67: Cell proliferation. VWF: Von Willebrand factor (micro-vessels). VIM: Neurofilament vimentin. PDGFR1: EMA and platelet-derived growth factor receptor 1. Scale bars: (**A**,**B**) 500 mm, (**D**,**E**) 75 mm. Abbreviations: patient-derived orthotopic xenograft (PDOX); Hematoxylin and eosin (H&E); magnetic resonance imaging (MRI); immunohistochemistry (IHC); epithelial membrane antigen (EMA).

**Figure 2 cancers-12-01478-f002:**
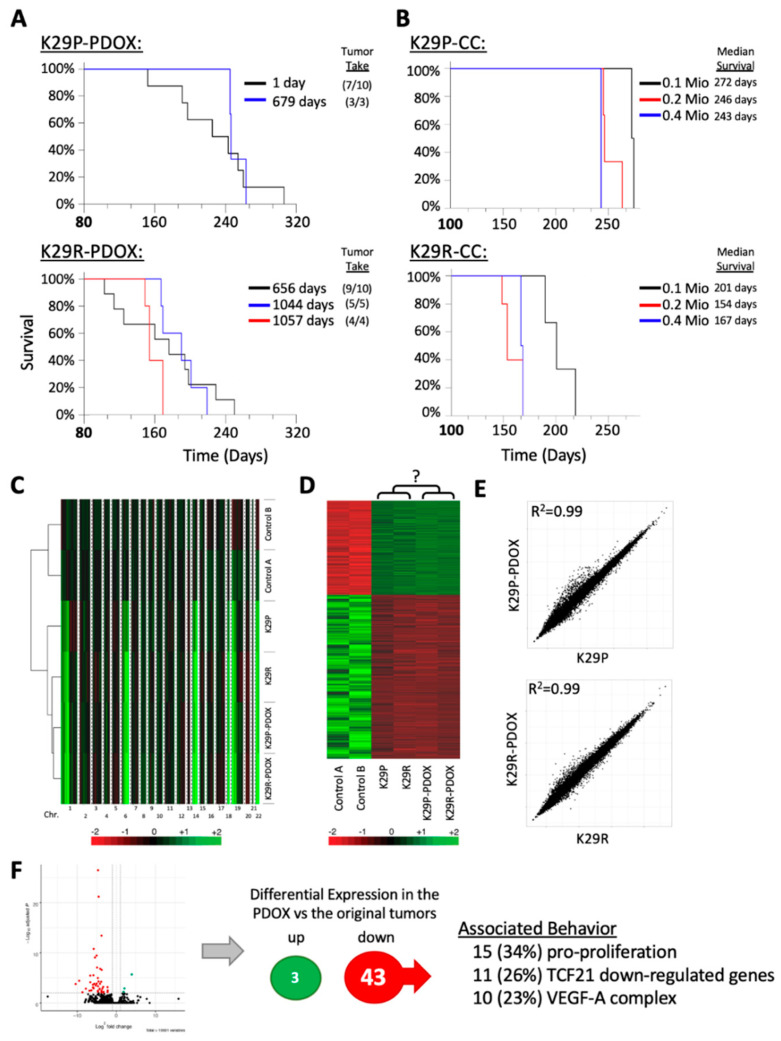
Tumorigenicity validation and molecular analysis. (**A**) Kaplan–Meier survival analyses from fresh and cryopreserved patient tumor cells (1 and 679 days after resection, respectively) as analyzed by log-rank analysis. Tumor take rate (xenograft formed/total mice implanted) are presented in parentheses. (**B**) Influence of initial number of implanted tumor cells, ranging from 0.l million (0.1 Mio) to 0.4 million (0.5 Mio), on animal survival. (**C**) Heatmap showing DNA copy number analysis of patient tumors and xenografts in comparison with two normal cerebral tissue samples (derived from diffuse intrinsic pontine glioma (DIPG) autopsy). (**D**) Heatmap showing the 2973 differentially expressed genes in the two PDOX models and the matching patient tumors using two normal cerebral tissues as references. (**E**) Pair-wise comparison of the original tumor gene expression profiles to the xenograph tumor gene expression profiles. The original and the PDOX expression signatures are highly similar. (**F**) Volcano-plot of differentially expressed genes between PDOX and original tumors. (Right side) Summary of the differentially expressed genes of the two xenographs compared to the original tumors (FDR ≤ 0.01) and their associated behavior. Abbreviations: patient derived orthotopic xenograft (PDOX); false discovery rate (FDR).

**Figure 3 cancers-12-01478-f003:**
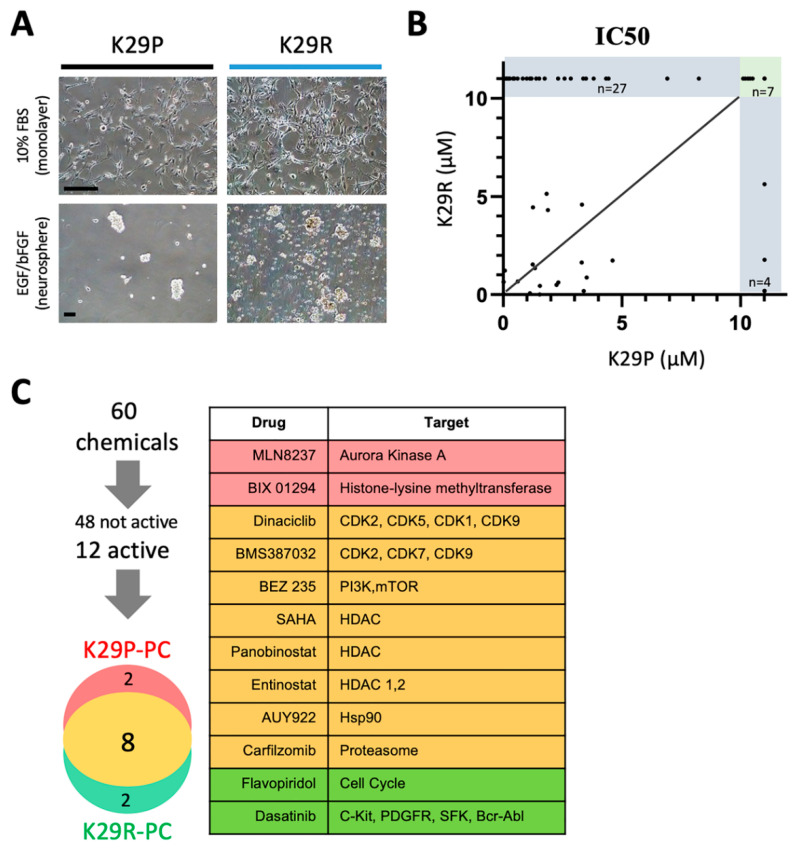
In vitro drug screening. (**A**) Growth of meningioma cells in vitro as monolayer (upper panel) or as tumorspheres (lower panel). (**B**) Comparison of drug responses (IC50) between the primary and the recurrent tumors. Among the 60 drugs tested, 38 drugs exhibited an IC50 greater than 10 µM (in the grey area) and majority of them (27/38 = 71%) were found in the recurrent tumor. (**C**) List of the 12 drugs that were found to be cytocidal (cell viability < 15% at 10 micromolar) in either the primary (K29P-PC) or the recurrent (K29R-PC) or both (*n* = 8). (**C**) Summary of the in vitro drug screen. Scale bars: 50 mm. Abbreviations: IC50, half maximal inhibitory concentration; mm, micrometers; PC, primary culture.

**Figure 4 cancers-12-01478-f004:**
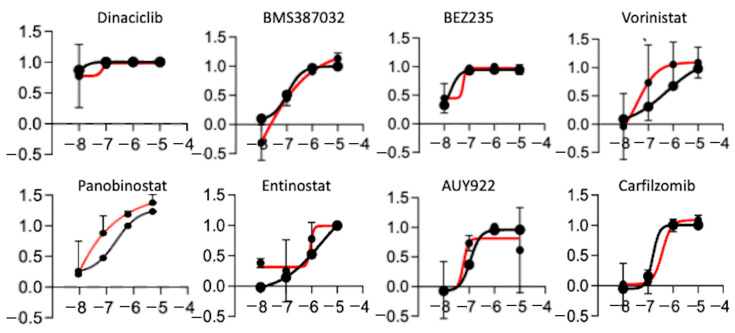
Drugs active in both primary cultures. Growth curve of the 8 drugs in the primary (K29P, black) and recurrent (K29R, red) patient’s primary culture.

**Figure 5 cancers-12-01478-f005:**
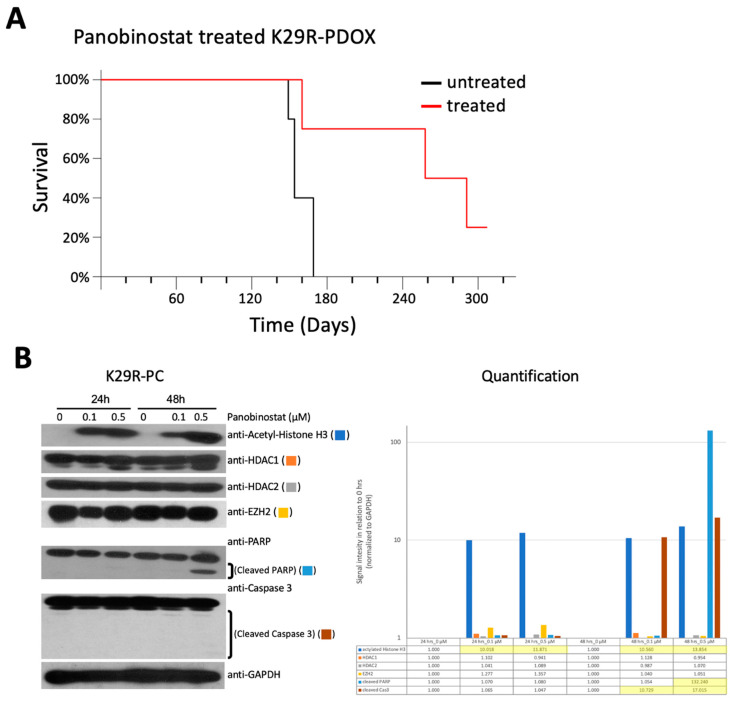
In vivo validation of Panobinostat. (**A**) In vivo validation of the anti-tumor activities of Panobinostat in the recurrent PDOX model (K29R-PDOX). (**B**). Western blot analysis showing the time and dose changes induced by Panobinostat in vitro in cultured meningioma cells derived from the recurrent tumor (K29R-PC) cells. Note the induction of acetylation in histone H3 (acetyle-H3) and cleavage of PARP in cells treated for 48 hr at 0.5 µM. Abbreviations: patient derived orthotopic xenograft (PDOX); primary culture (PC); Poly (ADP-ribose) polymerase (PARP).

**Table 1 cancers-12-01478-t001:** Summary of clinical information, tumor cell yield, and intracranial tumor formation of adult meningioma tumors.

Tumor ID		Age/Gender	WHO Grade	Type	Cell Tumorigenicity	
					Fresh	Cryo
K29	(P) rimary	72/F	Atypical (II)	C	7/10	3/3
	(R) ecurrent	72/F	Anaplastic (III)	C	1/4	9/10
K57	(P) rimary	65/M	Atypical (II)	C	0/10	-
K16	(P) rimary	71/F	Atypical (II)	C	0/12	-
	(R) ecurrent	72/F	Atypical (III) *	B	0/10	-
K18	(P) rimary	27/M	Atypical (II)	C	0/10	-
K20	(P) rimary	38/M	Atypical (II)	A	0/10	-
K21	(P) rimary	59/M	Classic (I)	Not sequenced	0/10	-
K26	(P) rimary	70/F	Classic (I)	A	0/5	-
K28	(P) rimary	33/F	Classic (I)	A	0/10	-
K30	(P) rimary	41/F	Classic (I)	B	0/5	-

* Atypical Meningioma with Rhabdoid Differentiation.

**Table 2 cancers-12-01478-t002:** List of anti-cancer drugs screened.

Drug Name	FDA Approved	Drug Name	FDA Approved
Barasertib		Trichostatin	
Bortezomib	yes	Cediranib	
Decitabine	yes	Dasatinib	yes
Enzastaurin		Panobinostat	
Lapatinib	yes	SAHA	yes
MLN8237		SNS-032	
NAC	yes	Carfilzomib	
Pazopanib	yes	Alvocidib	
Ponatinib		AUY922	
Quinacrine	yes	AZD-8931	
RO4929097		BEZ 235	
Ruxolitinib		Cabozantinib	
Selumetinib		Cilengitide	
SJ-172550		Dinaciclib	
Sodium butyrate	yes	Entinostat	
Sorafenib	yes	Fenretinide	
SP600125		Fostamatinib	
Temsirolimus	yes	Lenalidomide	
Veliparib		Nilotinib	
Vismodegib		PF 562271	
Zibotentan		Retinoic acid	
Crizotinib	yes	RO5045337	
GANT61		Saracatinib	
BMS 754807		SB-431542	
OSI-906		Tamoxifen	
MK-2206		Vandetanib	
Pelitinib		Vemurafenib	
BIX 01294		VER-155008	
Tozasertib		ABT 737	
Obatoclax		SJ-172550

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
