# Peer review of "Patient-Derived Orthotopic Xenograft (PDOX) Mouse Models of Primary and Recurrent Meningioma"

_cancers, 2020, doi:10.3390/cancers12061478_

Round 1

Reviewer 1 Report

Zhang et al present a paper on their efforts to create a PDOX model of meningiomas of different grades and molecular subtypes. As a clinical neuro-oncologist, I applaud this initiative for better models to study this very common, but understudied tumor type. The lack of efficient systemic therapies for recurrent (high-grade) meningiomas forms an urgent medical need.

The paper is well-written, and easy to read, even for a clinician. The detailed typing, and multi-platform-comparison, of the original tumors and the PDOX tumors are strong points, as well as the (seemingly) unbiased presentation of data.

My suggestions:

  • Abstract line 35-36: ‘Although typically benign, about 20% of these tumors recur despite surgery and radiation, and ultimately prove fatal.’. This suggests that overall disease-specific mortality for all meningioma patients is 20%. This might be changed to ‘…and may ultimately prove fatal.’.
  • Abstract line 55-56: ‘Yet the most malignant tumor type can be modeled,’. This wording seems to strong, since only the tumors of 1 patients were amenable to PODX modeling. As the authors themselves argue in the discussion (line 276-277): ‘It’s tempting to conclude that only Type C tumors are amenable to PDOX modeling—if it were not for the fact that three other Type C tumors (also mix of WHO grades II and III; Table 1) also failed to take.’. Consider rephrasing line 55-56.
  • Introduction line 64-65: ‘There are currently no effective therapies for patients with high-grade meningiomas.’. As the authors state, tumor control is achieved in a majority of these patients with available treatments (surgery and radiotherapy). Consider rephrasing e.g. ‘…for patients with surgery- and radiation-refractory high-grade meningiomas’.
  • Fig 1, line 110: please print the Hazard ratio and the exact p-value, not just ‘p<0.05’.
  • Results, lines 129-131: ‘Cell proliferation rate (Ki-67) was 2-5% in the primary tumor and xenografts but rose to ~10% in the recurrent tumors.’. Was the proliferation rate in the recurrent tumor xenografts also ~10%?
  • Figure 4, line ~238: there are typing errors in the drug names: ‘vorinistat’ should be ‘vorinostat’; ‘Panabinostat’ should be ‘Panobinostat’.

Author Response

We thank the reviewers for their thoughtful reading and constructive suggestions and questions for our manuscript entitled, “Patient-Derived Orthotopic Xenograft (PDOX) Mouse Models of Primary and Recurrent Meningioma”. We have been happy to amend the paper to accommodate their comments. For convenience, we quote the original reviewer comments in italicsbelow, and respond in regular font.

Reviewer 1

Zhang et al present a paper on their efforts to create a PDOX model of meningiomas of different grades and molecular subtypes. As a clinical neuro-oncologist, I applaud this initiative for better models to study this very common, but understudied tumor type. The lack of efficient systemic therapies for recurrent (high-grade) meningiomas forms an urgent medical need.

The paper is well-written, and easy to read, even for a clinician. The detailed typing, and multi-platformcomparison, of the original tumors and the PDOX tumors are strong points, as well as the (seemingly) unbiased presentation of data.

            We’re gratified that the reviewer found the paper clear.

My suggestions:

Abstract line 35-36: ‘Although typically benign, about 20% of these tumors recur despite surgery and radiation, and ultimately prove fatal.’. This suggests that overall disease-specific mortality for all meningioma patients is 20%. This might be changed to ‘…and may ultimately prove fatal.’

            We have edited the sentence as requested.

Abstract line 55-56: ‘Yet the most malignant tumor type can be modeled,’. This wording seems to strong, since only the tumors of 1 patients were amenable to PODX modeling. As the authors themselves argue in the discussion (line 276-277): ‘It’s tempting to conclude that only Type C tumors are amenable to PDOX modeling—if it were not for the fact that three other Type C tumors (also mix of WHO grades II and III; Table 1) also failed to take.’. Consider rephrasing line 55-56.

            Very good point. Although we wrote this with a mental italics over “can”, suggesting possibility rather than certainty, we have rephrased to ensure readers understand the difficulty: “Yet at least some of the most malignant tumors caxn be modeled...”

Introduction line 64-65: ‘There are currently no effective therapies for patients with high-grade meningiomas.’. As the authors state, tumor control is achieved in a majority of these patients with available treatments (surgery and radiotherapy). Consider rephrasing e.g. ‘…for patients with surgery- and radiation-refractory high-grade meningiomas’.

            Very good suggestion; we have edited the sentence accordingly.

Fig 1, line 110: please print the Hazard ratio and the exact p-value, not just ‘p<0.05’.

            Done: the hazard ratio is 2.474 and the p-value P=0.0390

Results, lines 129-131: ‘Cell proliferation rate (Ki-67) was 2-5% in the primary tumor and xenografts but rose to ~10% in the recurrent tumors.’. Was the proliferation rate in the recurrent tumor xenografts also ~10%?

            Yes, we have adjusted the sentence to reflect that.

Figure 4, line ~238: there are typing errors in the drug names: ‘vorinistat’ should be ‘vorinostat’; ‘Panabinostat’ should be ‘Panobinostat’.

            Our apologies. Thanks for pointing this out; corrected.

Reviewer 2 Report

The authors aimed to develop patient-derived orthotopic xenograft (PDOX) mouse models of human meningioma.  Only the most malignant tumor type could be modeled from tumor cells derived from one patient. The cryopreserved primary tumor cells were used for preclinical drug testing. The HDAC inhibitor Panobinostat in K29R-PDOX mice prolonged mouse survival.

PDOX modeling could not be achieved with tumor cells from the other patients with grade II or III meningioma.

Could the present model be reproducible with other grade II or III meningiomas, or was there something particular in the biology of the tumor of this patient to allow PDOX modeling?

If the results were reproducible, grade II and III meningiomas would mostly benefit from drug trials.

Author Response

We thank the reviewers for their thoughtful reading and constructive suggestions and questions for our manuscript entitled, “Patient-Derived Orthotopic Xenograft (PDOX) Mouse Models of Primary and Recurrent Meningioma”. We have been happy to amend the paper to accommodate their comments. For convenience, we quote the original reviewer comments in italicsbelow, and respond in regular font.

Reviewer 2

The authors aimed to develop patient-derived orthotopic xenograft (PDOX) mouse models of human meningioma. Only the most malignant tumor type could be modeled from tumor cells derived from one patient. The cryopreserved primary tumor cells were used for preclinical drug testing. The HDAC inhibitor Panobinostat in K29R-PDOX mice prolonged mouse survival. PDOX modeling could not be achieved with tumor cells from the other patients with grade II or III meningioma. Could the present model be reproducible with other grade II or III meningiomas, or was there something particular in the biology of the tumor of this patient to allow PDOX modeling?

            We agree, this is the million-dollar question. We hope to conduct a systematic study to understand why meningiomas are so recalcitrant to modeling, but it will take quite some time (and considerable funds) to really answer this question.

If the results were reproducible, grade II and III meningiomas would mostly benefit from drug trials.

            We agree that reproducibility is key. We hope this work spurs interest from other groups in developing better ways to model meningioma.